# Reconstructing 3D subsurface salt flow

Stefan Back[1], Sebastian Amberg[2,3], Victoria Sachse[3,4], and Ralf Littke[3]

[1] Tectonics and Geodynamics, EMR, RWTH Aachen University, Germany
[2] Geological Institute, EMR, RWTH Aachen University, Germany
[3] Institute of Geology and Geochemistry of Petroleum and Coal, EMR, RWTH Aachen University, Germany
[4] Forschungszentrum Jülich GmbH, Projektträger Jülich, Germany

*Correspondence to:* Stefan Back (stefan.back@emr.rwth-aachen.de)

**Abstract.** Archimedes' principle states that the upward buoyant force exerted on a solid immersed in a fluid is equal to the weight of the fluid that the solid displaces. In this 3D salt-reconstruction study we treat Zechstein evaporites in the Netherlands as a pseudo-fluid with a density of 2.2 $g/cm^3$, overlain by a lighter and solid overburden. 3D sequential removal (backstripping) of a differential sediment load above the Zechstein evaporites is used to incrementally restore the top Zechstein surface. Assumption of a constant subsurface evaporite volume enables the stepwise reconstruction of base Zechstein and the approximation of 3D salt-thickness change and lateral salt re-distribution over time.

The salt restoration presented is sensitive to any overburden thickness change caused by tectonics, basin tilt, erosion or sedimentary process. Sequential analysis of lateral subsurface salt loss and gain through time based on Zechstein isopach difference maps provides new basin-scale insights into 3D subsurface salt flow and redistribution, supra-salt depocentre development, the rise and fall of salt structures, and external forces' impact on subsurface salt movement. The 3D reconstruction procedure is radically different from classic backstripping in limiting palinspastic restoration to the salt overburden; followed by volume-constant balancing of the salt substratum. The unloading approach can serve as a template for analysing other salt basins worldwide and provides a stepping stone to physically sound fluid-dynamic models of salt tectonic provinces.

## 1 Introduction

Archimedes (c. 246 BC) proposed - in short - that the upward buoyant force exerted on a solid body immersed in a fluid, whether fully or partially submerged, is equal to the weight of the fluid that the solid body displaces. This principle is an

essential law of physics and fluid mechanics. In geoscience, it forms e.g. the foundation of Airy Isostasy (Airy, 1855). This study uses Archimedes' principle to reconstruct 3D subsurface salt flow through geological time by treating salt as a dense fluid phase ($\rho$ = 2.2 g/cm$^3$) in which lighter overburden rocks (solids) float (Fig. 1a).

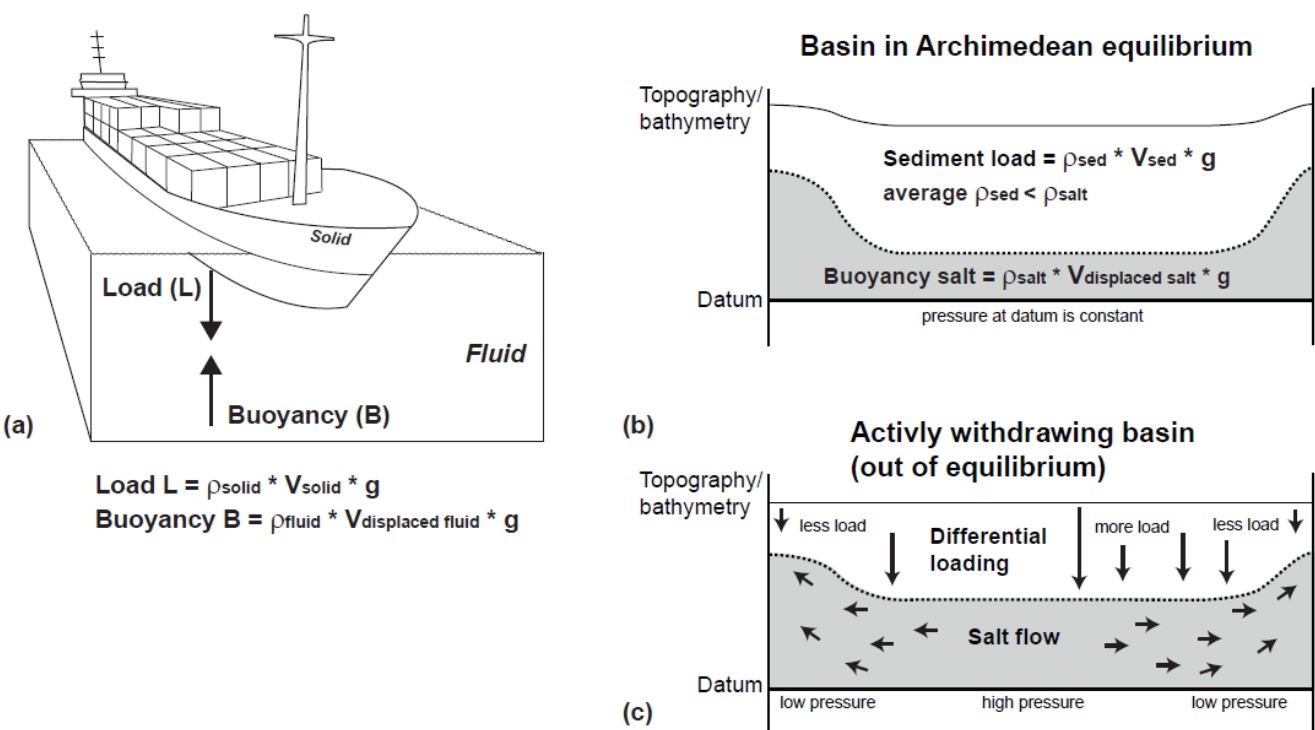

**Figure 1.** Archimedes' principle. (a) Ship loaded and unloaded floating on water. Buoyancy as upward force exerted by the fluid (water) that opposes the weight of the partially immersed ship. (b) Basin above subsurface evaporites in Achimedean equilibrium. Salt treated as dense pseudo-fluid (= 2.2 g/cm$^3$) loaded by cumulatively lighter overburden rocks (solid). Backstripping corresponds to incremental unloading of the overburden (sensu Maystrenko et al. 2013). Archimedean restoration of the Zechstein Basin example justified by slow subsurface salt flow. c) Basin out of equilibrium by major differential loading with significant salt flow and high differential pressure at datum level.

The buoyancy driver for subsurface salt movement was already proposed over 100 years ago by Aarhenius and Lachmann (1912) and subsequently formalised by Barton (1933) and Nettleton (1934). Trusheim (1957, 1960) was a major proponent of this theory, and applied this approach of analysing salt flow to the NW European salt basin. In an early study on

potential nuclear-waste storage sites, Kehle (1980) specified that "sediment loading, not buoyancy, sensu stricto", drives subsurface salt movement. Kehle (1988) pointed out several weaknesses in the original buoyancy theory mainly from a hydrodynamic perspective. He emphasized two main controls for salt flow, gravity head and pressure head, and stressed the importance of differential loading (resulting in high fluid-head gradients) for subsurface salt movement. Waltham (1997) quantitatively investigated non-buoyant causes of salt movement (compression causing overburden thickening; flexural overburden buckling; drag) and compared their effectiveness to buoyancy.

Few rocks behave as close as a Newtonian fluid as rock salt (e.g. Van Keken et al., 1993; Davison et al., 1996; Koyi, 2001; Gemmer et al., 2004; Hudec and Jackson, 2007; Jackson and Hudec, 2017). Various modelling studies have consequently treated subsurface salt as a pseudo-fluid flowing in the subsurface considering its sedimentary overburden as solid (e.g. Jackson and Vendeville, 1994; Maystrenko et al., 2013). The assumption of the supra-salt stratigraphy floating on a thick subsurface salt layer requires salt, similar to a viscous fluid, to be in Archimedean (hydrostatic) equilibrium with the overburden (Fig. 1b). In such balance, the depletion of the subsurface salt layer will either be a passive response to differential loading by supra-salt sediments, or create ("actively", e.g. by dissolution) additional accommodation space to be loaded by sediment. In turn, thickening of the subsurface salt layer will either actively destroy supra-salt accommodation space or passively respond to an externally forced decrease of the salt overburden (e.g. localized erosion). The Archimedean equilibrium approach with a solid overburden floating on an evaporite layer is supported by the observation that salt flows when loaded, and that faulting rather than folding characterises deformation in the overburden (Davison 2009; Warren, 2016). However, it must be recognized that all actively withdrawing basins are to some degree out-of-equilibrium (Fig. 1c). Yet, rather slow and local lateral salt flow, such as documented over large areas at various time intervals in this Zechstein Basin case study, can be seen as supporting a static Archimedean approach for salt reconstruction (Fig. 1b). The applicability of this method in salt provinces characterised by rapid sediment accumulation, fast salt movement, major basin subsidence and/or the occurrence of large allochthonous salt bodies (e.g. Gulf of Mexico: Duffy et al., 2019; Santos Basin: Jackson et al., 2015) has yet to be studied.

In this 3D backstripping exercise we work around the complications of "fluid-dynamic" modelling of subsurface salt in that we simply measure 3D change in space through time (at Top salt) rather than simulating details of a salt-flow regime. Elementary backstripping theory proved sufficient to determine areas of accommodation loss and gain through time by

overburden restoration, no matter what the exact flow properties of salt or the cause for loss and gain of depositional space (e.g. tectonics, differential sedimentation, erosion). The backstripping and buoyancy compensation results presented are valid with the provision that salt flows into surplus space (salt gain), if available; that subsurface evaporites will laterally move and

70 redistribute when differentially loaded (salt loss); and that the subsurface salt volume remains constant. The analysis of lateral subsurface salt loss and gain through time provides basin-scale insights into 3D subsurface salt movement, the development of supra-salt depocentres, the growth and decay of salt structures, and external forces' impact on salt systems. The 3D reconstruction procedure presented is mathematically easy and computationally quick. Various 1D, 2D, and 3D unloading and isostatic balancing algorithms for retracing the reconstruction approach are readily available in several free and commercial

geological interpretation and modelling software.

## 2 Study area, data and methods

The Permian Zechstein Group in the subsurface of the Netherlands, Central Europe (Fig. 2a) accumulated in the foreland of the Variscan orogen (Geluk 2007). The Zechstein Group of the onshore Netherlands comprises five evaporite

cycles (Z1-Z5; Van Adrichem Boogaert and Kouwe, 1993; Geluk, 2007) with several hundreds of metres of rock salt and anhydrite deposited mainly in Z2 and Z3 (Fig. 2b). Several small tectonic pulses are reported to have occurred during Zechstein times, with partly extensional and partly compressional faulting mainly affecting anhydrite platforms at the Zechstein Basin margins (Geluk, 1999). The occurrence of Zechstein evaporites in the Netherlands' subsurface influenced the post-Permian geological development. The visco-plastic behaviour of salt under loading and compressive tectonic stress (Remmelts, 1995)

led to the development of numerous salt structures, mainly salt rollers, salt anticlines and salt walls (e.g. Fig. 3). Many of these structures were not actively diapiric and did not grow further when buried (e.g. Trusheim, 1963).

The Zechstein Group of the onshore Netherlands is overlain by the Lower and Upper Germanic Trias groups (RB, RN), the Jurassic Altena (AT), Schieland (SL), Scruff (SG) and Niedersachsen (SK) groups, the Cretaceous Rijnland (KN) and Chalk groups (CK), the Paleogene Lower (NL) and Middle (MN) North Sea groups and the Neogene to recent Upper

North Sea Group (NU; Figs. 2b; 3a, b; TNO-NITG, 2004; Duin et al., 2006; Wong et al., 2007). For simplicity, this study

treats the entire Permian Zechstein Group as one evaporite unit reacting to loading and unloading over geological time as a Newtonian fluid.

Seven 3D surfaces in depth (m) from the public 3D regional subsurface layer model of the Netherlands "DGM-deep v5" (TNO-NLOG, 2022) form the study's stratigraphic framework. Horizon and lithology data are from the NLOG and DINOLoket public databases. Fifteen individual 3D seismic-reflection volumes (in two-way-time (TWT); two volumes additionally pre-stack depth migrated) were used for the identification of key structural elements, subsurface salt occurrence, unconformities and overburden stratigraphy (Fig. 2). Conversion of subsurface data from time (ms TWT) to depth (m) and vice versa was based on the velocity model of Van Dalfsen et al. (2006).

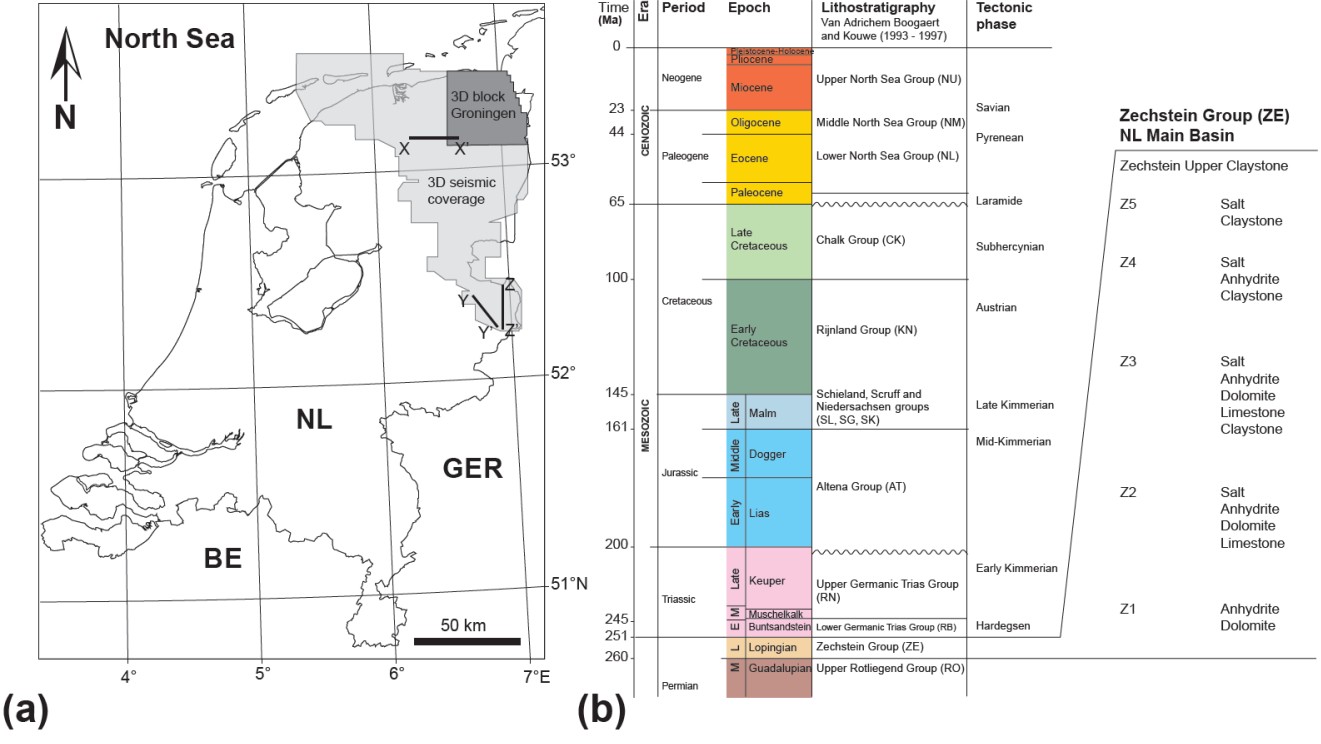

**(a)**

**(b)**

**Figure 2.** (a) Study area in the NE of the Netherlands and 3D seismic coverage. 3D block Groningen used for quality control. Lines X-X' and Y-Y' shown in Figure 3. BE = Belgium; GER = Germany; NL = The Netherlands. (b) Stratigraphy of the study area (after Van Adrichem Boogaert and Kouwe, 1993). Detailed lithostratigraphic subdivision of the Zechstein Group on the right. Stratigraphic abbreviations used on following figures.

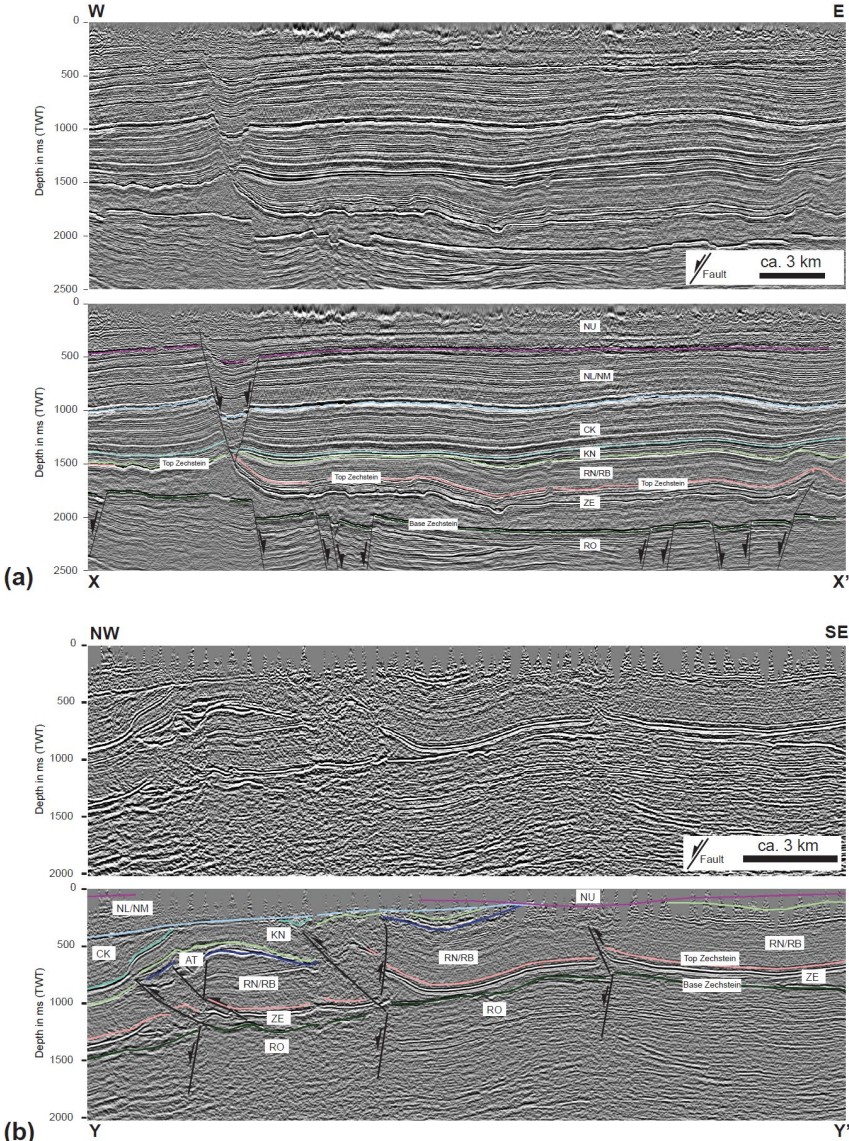

**Figure 3.** (a) Seismic-reflection line X-X' across the northern-central study area. Top – uninterpreted, base – interpreted. Note Zechstein unit (ZE) and bright, strong amplitude reflection near top imaging partly deformed and folded intra-salt Zechstein 3 stringer (Strozyk et al., 2012). Incremental restoration (see e.g. Figs. 4, 5) documents that most salt rollers, anticlines and salt walls did not grow further when buried; instead, many early salt-cored highs experienced salt loss through time. (b) Seismic-reflection line Y-Y' across southern study area. Top – uninterpreted, base – interpreted. Note lack of upper Mesozoic and Cenozoic Zechstein overburden in the south. For line locations and stratigraphic abbreviations see Figure 2.

Backstripping for stratigraphic restoration and salt-flow monitoring was initially applied to individual 3D seismic-reflection volumes (a.o. 3D block Groningen; Fig. 2a). The method proved simple, quick and effective and was therefore

immediately extended to the entire (ca. 10,000 km$^2$) NE Netherlands. Strata above the Zechstein were assigned average lithologies (Table 1) with the definition of average rock type (shale, sand, chalk, shaley sand), compaction trends and density/depth relationships taken from the North Sea database of Sclater and Christie (1980) without modification. Young's Modulus and Poisson Ratio data are from Hunfeld et al. (2021). In all cases the present-day cumulative average density of the column of vertical overburden (= grain density + porosity; pores filled with water) was lighter than the density of the evaporite

substratum (fluid with $\rho = 2.2$ g/cm$^3$), and should have been so in the past.

**Table 1:** Model rock properties based on stratigraphy and rock type (Sclater and Christie, 1980; Hunfeld et al., 2021). Average rock type always assigned as 100% of unit. All pores assumed to have been filled with water.

| Lithostratigraphic Unit | Average Rock Type | Initial Porosity (%) | Depth Coefficient (km$^{-1}$) | Grain density g/cm$^3$ | Young's Modulus (MPa) | Poisson Ratio |
|---|---|---|---|---|---|---|
| Upper North Sea Group | Sand | 0.49 | 0.27 | 2.65 | 15000 | 0.29 |
| Middle and Lower North Sea Group | Sand | 0.49 | 0.27 | 2.65 | 15000 | 0.29 |
| Chalk Group | Chalk | 0.70 | 0.71 | 2.71 | 37500 | 0.32 |
| Rijnland Group | Shaley sand | 0.56 | 0.39 | 2.68 | 23750 | 0.3 |
| Schieland, Scruff and Niedersachsen Groups | Shaley sand | 0.56 | 0.39 | 2.68 | 23750 | 0.3 |
| Altena Group | Shale | 0.63 | 0.51 | 2.72 | 32500 | 0.3 |
| Germanic Trias Group | Shale | 0.63 | 0.51 | 2.72 | 32500 | 0.3 |
| Zechstein Group | Shale | unbalanced | unbalanced | 2.2 | unbalanced | unbalanced |


The backstripping observation that the cumulative average overburden density remained in the study area always lower than that of the Zechstein Group might be surprising, as every sediment will become at some depth denser than salt. Yet, in the study area the depth of Top Zechstein was never very great (in most areas < 2500 m; see Fig. 3 for present-day situation).

Since i) both the Chalk Group and the North Sea Group were never deeply buried; ii) both groups constitute the main part of the overburden (Fig. 3); and iii) chalk can preserve very high porosities at depth (30-50% at ca. 2500m in the North Sea example of Sclater and Christie, 1980), we estimate that in the study area over 3000 m of sedimentary cover with a significant shale content would be needed to attain a cumulative average overburden density exceeding 2.2 g/cm$^3$.

The reconstruction approach forwarded in this study differs from standard backstripping (e.g. Rowan, 2003;
Maystrenko et al., 2013; Turcotte and Schubert, 2014) in that it does not apply any vertical shear restoration. After each unloading workstep a new model top surface was calculated by leaving the remaining overburden float on salt. As result there is a residual topography on each restored top surface. Figure 4 illustrates the general 3D reconstruction methodology applied, Figure 5 shows exemplarily a complete restoration sequence between the present-day and the Base Triassic (251 Ma) along 2D sections NORG XL8000 (line X-X') and TWENTE IL 9000 (line Z-Z').

The present-day structural framework formed the base for all restorations (Figs. 4-6). The first restoration step (e.g. Fig. 4a) removed the uppermost stratigraphic layer. As result from unloading by stratal removal, the remaining stratigraphic column down to Top salt was readjusted by Airy-type vertical unloading (buoyancy compensation). Remnant space above the top pile of sediment up to sea level was then filled with seawater (e.g. Fig. 4b). Zechstein evaporites and any surface and unit below were excluded from unloading. Restoration step 2 then shifted the remaining, unbalanced Base Zechstein 3D surface
vertically upward into a new position (e.g. Fig. 4c) constrained by keeping the subsurface Zechstein volume in the 3D framework model at its initial value (full study area ca. 6.38 x 10$^{12}$ m$^3$). Zechstein thickness was then measured across the study area and plotted as isopach map (Fig. 6). Unloading (restoration step 1), volumetric salt-balancing (restoration step 2) and salt-thickness measurement were recurrently repeated (see restoration steps 3 and 4; Figs. 4d and 4e) until the entire salt overburden was removed (e.g. Fig. 5).


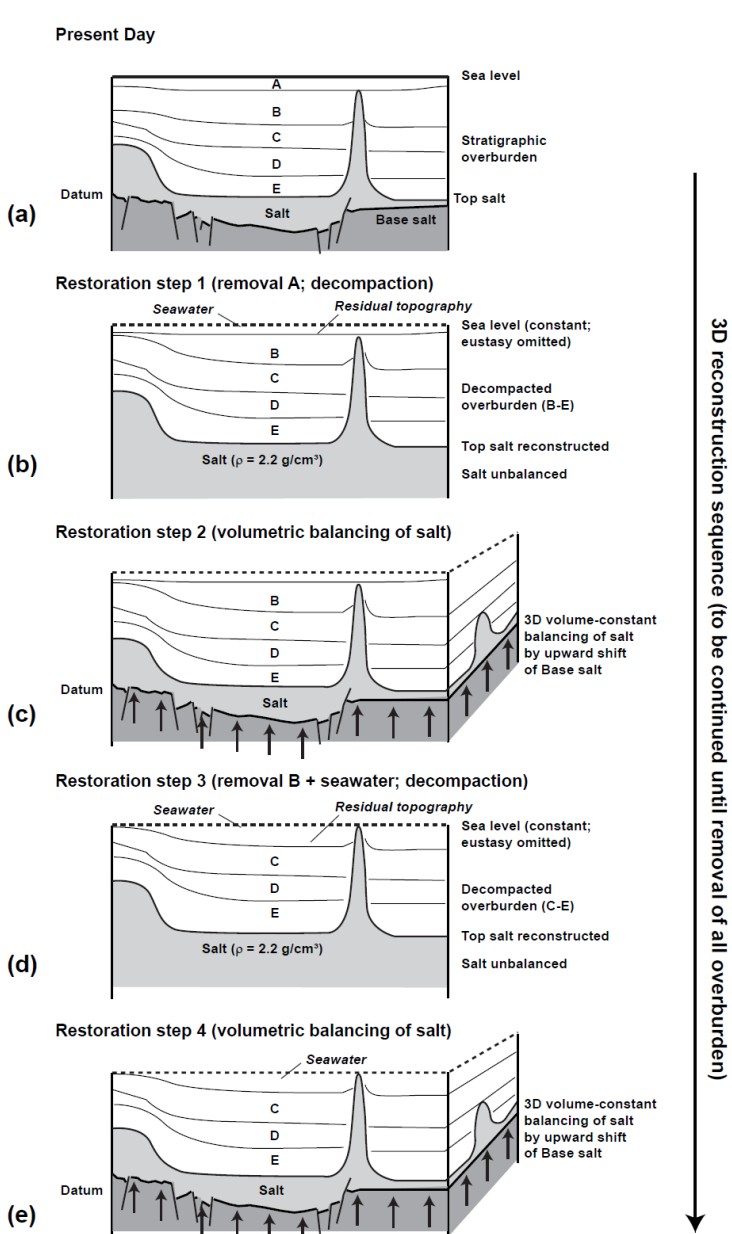

**Figure 4.** Restoration methodology. (a) Present-day situation. (b) Restoration step 1: Removal of top layer A. Airy-type vertical unloading of sedimentary column down to Top salt. Decompaction of sedimentary column down to Top salt (only restoration scenarios 2 and 3). Salt unit remains unbalanced. Residual topography flooded with seawater. (c) Restoration step 2: Upward vertical shift of unbalanced Base salt into new position constrained by keeping salt volume constant. (d) Same as restoration step 1 but removal of new top layer B. (e) Same as restoration step 2 with remaining stratigraphy. Entire 3D unloading procedure to be continued until removal of all salt overburden (Fig. 5).

The sensitivity of the 3D Zechstein-thickness reconstructions to varying backstripping parameters was tested using three different restoration scenarios. In restoration scenario 1, buoyancy compensation was local ("Airy isostasy" above salt), i.e. only vertically below the load, and decompaction was omitted from backstripping to produce the simplest reconstruction

(Fig. 6a). In restoration scenario 2 (Figs. 5, 6b), buoyancy compensation was kept local ("Airy type") but decompaction of the overburden was included in backstripping. Restoration scenario 3 used flexural balancing instead of "Airy type" unloading in order to account for the cohesive strength of the overburden, and included decompaction (Fig. 6c). In scenario 3 every restoration step used the respective average overburden thickness calculated during the preceding workstep to define a new individual effective elastic thickness (Te) above salt. Irrespective of the scenario applied, after every unloading step the

evaporite volume below the backstripped top Zechstein surface was readjusted and restored to the initial model volume (ca. $6.38 \times 10^{12}$ m$^3$) by shifting the base Zechstein surface upwards. The geometry (external form) of the Zechstein base and the Zechstein volume were kept constant in all reconstruction steps.

Identical salt-thickness restoration results could have been achieved by moving the top Zechstein and overburden downwards to keep the salt volume constant. This indicates that the restoration methodology used is independent of a reference

datum and consequently does not support referenced surface-topography analysis or palaeo-geographic reconstruction. The restoration approach in its current form is limited to incrementally backstrip the shallow post-salt overburden for the sole purpose of 3D true-to-volume reconstruction of the salt substratum, explicity excluding isostatic balancing of the crust-mantle equilibrium. The method therefore cannot be compared with classic crustal backstripping of salt systems (e.g. sensu Rowan, 2003; basic principles in Turcotte and Schubert, 2014).

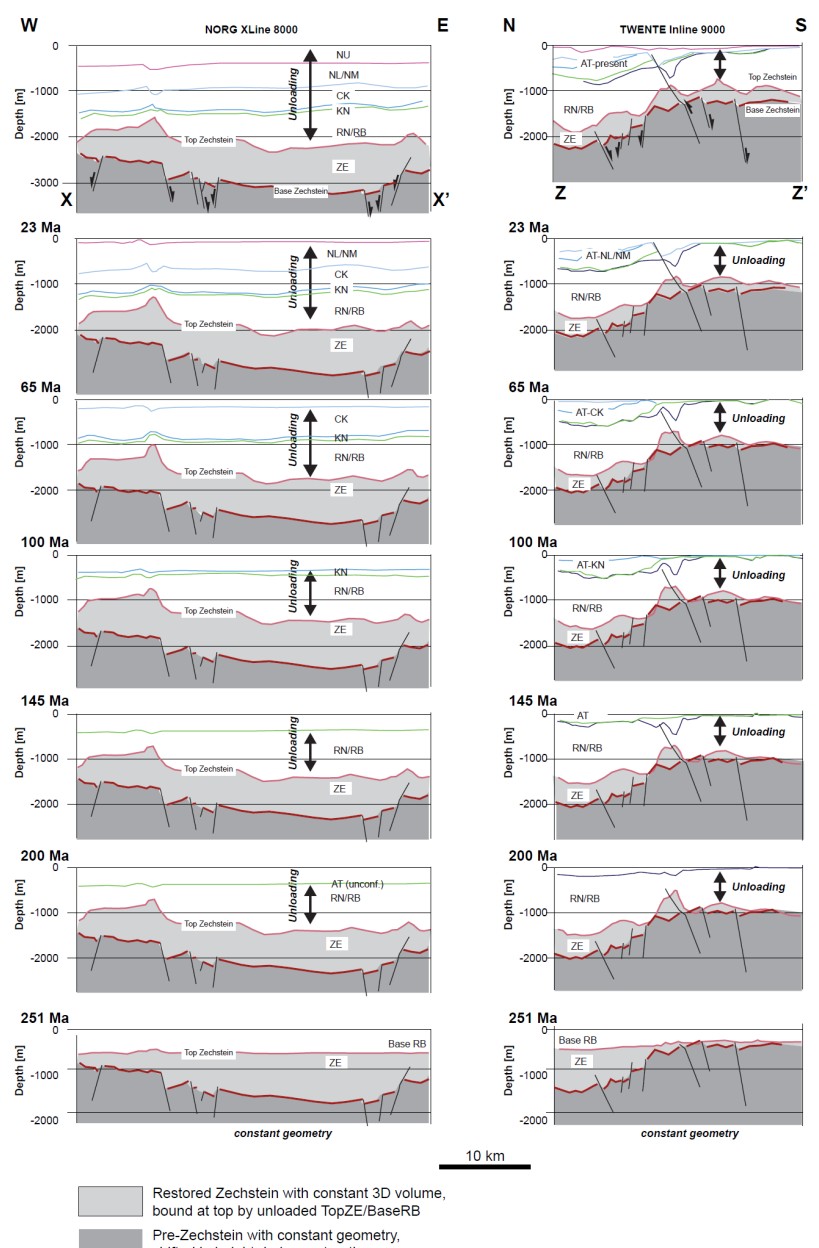


**Figure 5.** Restoration examples (scenario 2 - unloading and decompaction). Selected geological sections (X-X' = NORG XL 8000; Z-Z' = TWENTE IL 9000) through the 3D model that illustrate the sequential evolution of structure, stratigraphy and thickness of the Zechstein unit. Note absence of Jurassic strata (AT unconformity) along cross section X-X'. Also note pronounced flattening of Top salt towards 251 Ma. For illustration of 3D salt-thickness change and 3D subsurface salt flow through time see Figures 6 to 8. For location of sections see

Figures 2 and 6-10).

## 3 Salt thickness reconstruction and salt loss-gain plots

True-to-volume Zechstein unloading in six time steps of ca. 25 - 50 Myrs duration restored the 3D subsurface evaporite thickness and distribution between today and 251 Ma (Fig. 6). Key differences between the three example scenarios are the omission (Fig. 6A) versus the inclusion of overburden decompaction during backstripping (Figs. 6b, c); and the use of pure vertical unloading ("Airy unloading"; Figs. 6a and b) versus flexural overburden balancing (Fig. 6c). At first sight, Zechstein isopachs between today and 200 Ma appear in all reconstructions relatively similar. A significant difference characterises all three restoration scenarios in the interval between 200 and 251 Ma. At 251 Ma, all reconstructions restore major Zechstein thickness maxima (>1.5 km) in the Lower Saxony Basin (LSB) and in the northern Lauwerszee Trough (LT), irrespective of the reconstruction approach used (Fig. 6). Few isolated thickness maxima remain more or less fixed at all times in restoration scenarios 1 and 2 that apply pure vertical unloading (Figs. 6a, b). These maxima correspond to piercement salt domes (e.g. Pieterburen, Winschoten) that remain unbalanced due to the lack of a vertical overburden. Such unbalanced salt structures account for less than 5% of the total Zechstein model volume in scenarios 1 and 2 (Figs. 6a, b). In restoration scenario 3, salt piercement structures change their shape during reconstruction due to overburden flexure affecting neighbouring areas.

In contrast to the rather subtle differences in Zechstein isopachs (Fig. 6), the difference plots between successive pairs of isopach maps (Figs. 7 and 8) show considerable variation. Salt loss and gain is local and represents lateral flow of salt within the model. Loss represents salt withdrawal and lateral expulsion; gain represents local salt inflation by salt influx. Salt loss and gain in the range of of several hundreds of metres to >1 km are highest in all restoration scenarios in the Triassic (251 - 200 Ma); at this time, major salt loss characterises the LSB, less salt loss the northern LT (Fig. 7). In all restoration scenarios salt escape is mainly to the Friesland Platform (FP) and Groningen High (GH), both gaining between ca. 500 m (Scenario 2, Fig. 7b) and 900 m (Scenario 3, Fig. 7c) of evaporites. The Jurassic (200-145 Ma) difference maps display uniform salt gain across most of the study area due to the presence of the Base Cretaceous unconformity. The LSB shows in this time subsurface salt loss between 150 m (Scenario 1, Fig. 7a) and 300 m (Scenario 3, Fig. 7c); the FP shows salt loss between 30 and 90 m (Fig. 7).

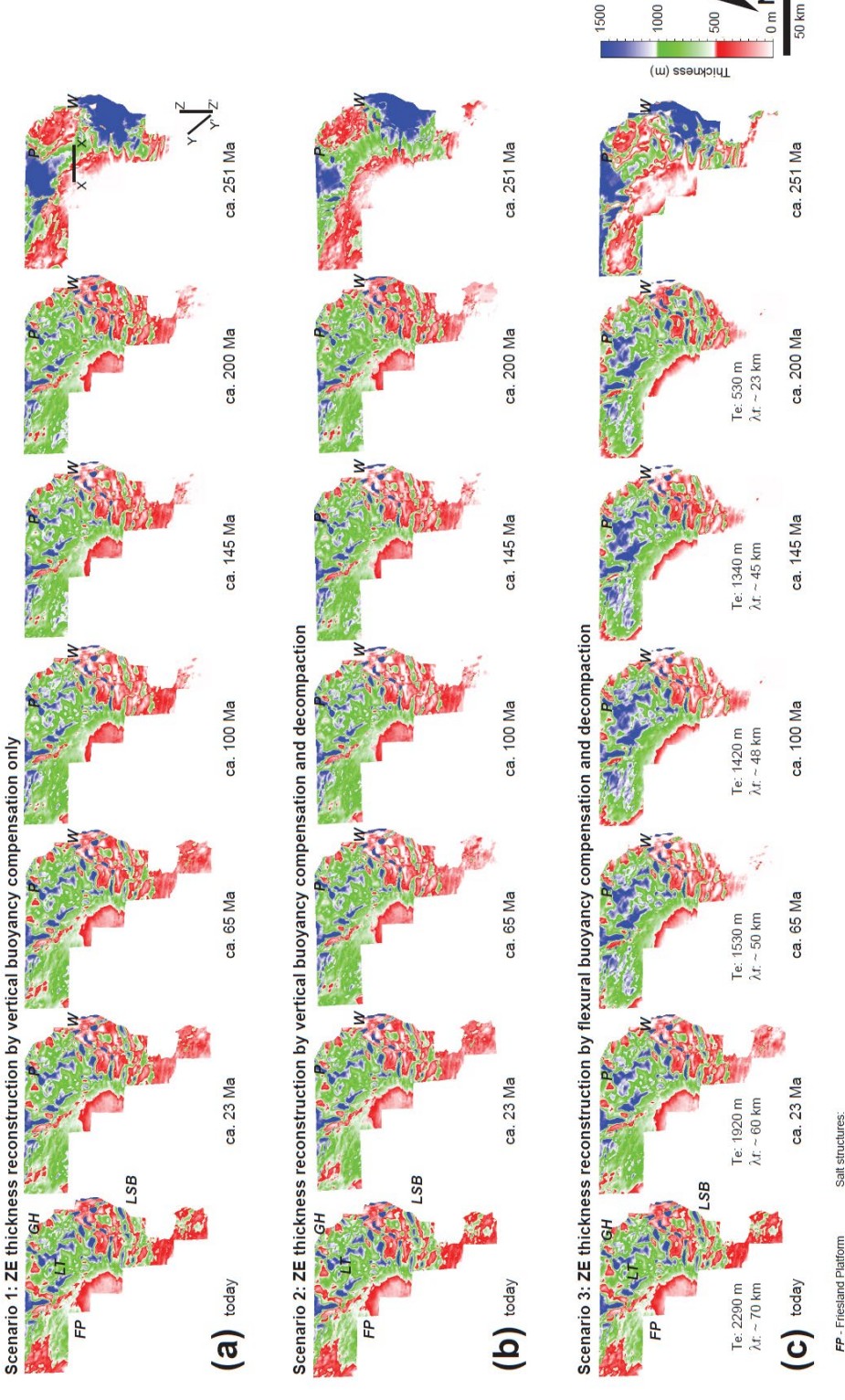

**Figure 6.** 3D Zechstein thickness-reconstruction results by backstripping. Note 251 Ma thickness maximum of Zechstein in Lower Saxony Basin (LSB) and Lauwerszee Trough (LT) in all reconstructions. (a) Backstripping scenario 1 – restored Zechstein thicknesses by vertical buoyancy compensation ("Airy balancing") omitting decompaction. (b) Backstripping scenario 2 - restored Zechstein thicknesses by vertical buoyancy compensation ("Airy balancing") including decompaction. (c) Backstripping scenario 3 - restored Zechstein thicknesses by flexural buoyancy compensation including decompaction. Effective elastic thickness (Te) of the overburden calculated as average overburden thickness from preceding restoration step; note corresponding change of flexural wavelength (f) during backstripping. All reconstructions using submarine conditions. FP = Friesland Platform; GH = Groningen High; P = Pieterburen; W = Winschoten. 2D restoration (scenario 2) extracted along sections X-X' and Z-Z' (see (a), 251 Ma) shown in Figure 5.

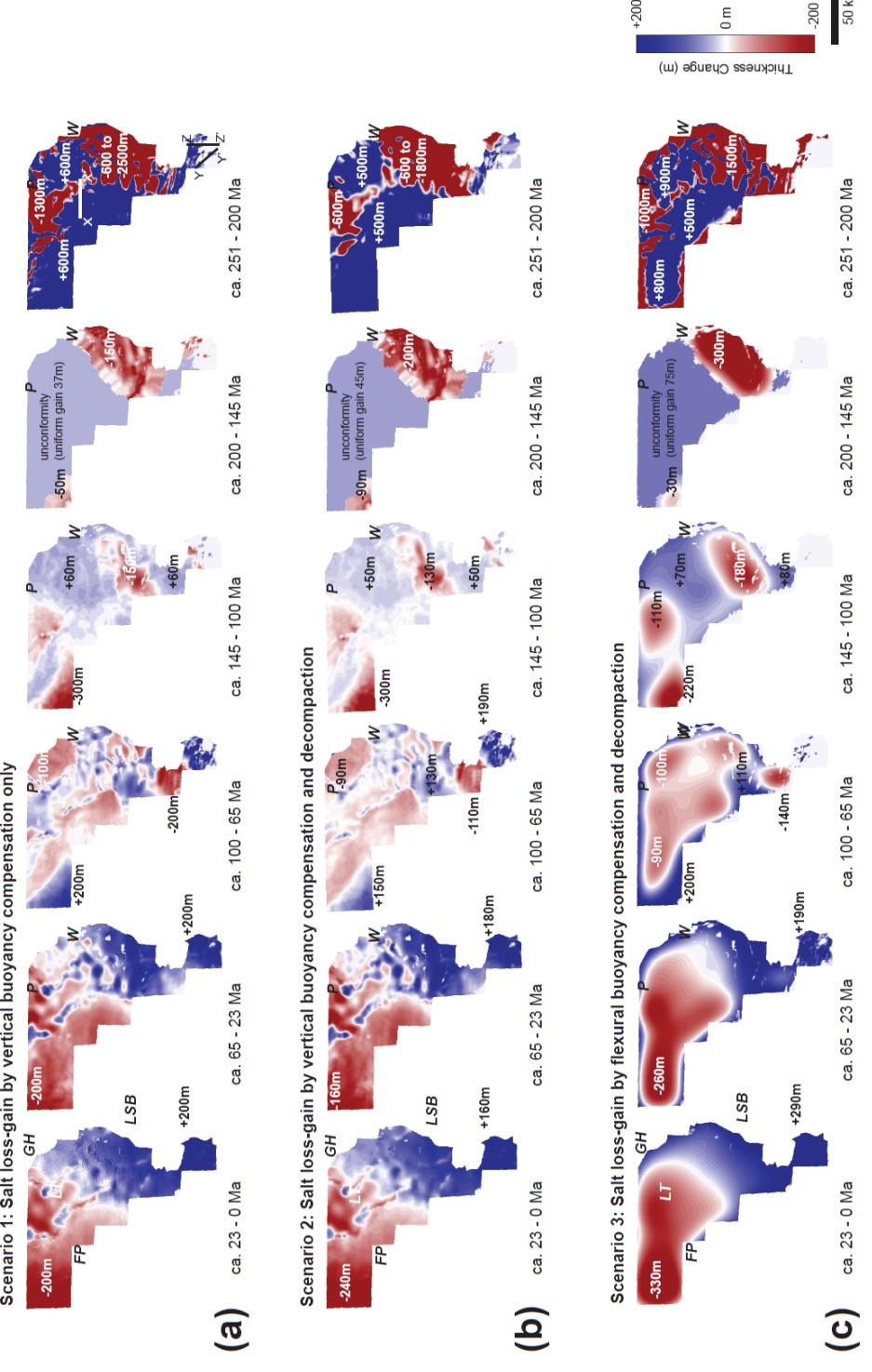

**Figure 7:** Differences calculated between successive pairs of isopachs of Figure 6. (a) Salt loss-gain plot of backstripping scenario 1. (b) Salt loss-gain plot of backstripping scenario 2. (c) Salt loss-gain plot of backstripping scenario 3. Note similarity between vertical buoyancy compensation of backstripping scenarios 1 and 2 documenting limited significance of decompaction. Note pronounced difference between flexural buoyancy compensation (c) and vertical balancing (a, b) between recent time and the Early Cretaceous (145-100 Ma). Late Cretaceous salt gain pronounced in salt structures above LT boundary faults. 2D restoration (scenario 2) extracted along sections X-X' and Z-Z' (see (a), 251 Ma) shown in Figure 5.

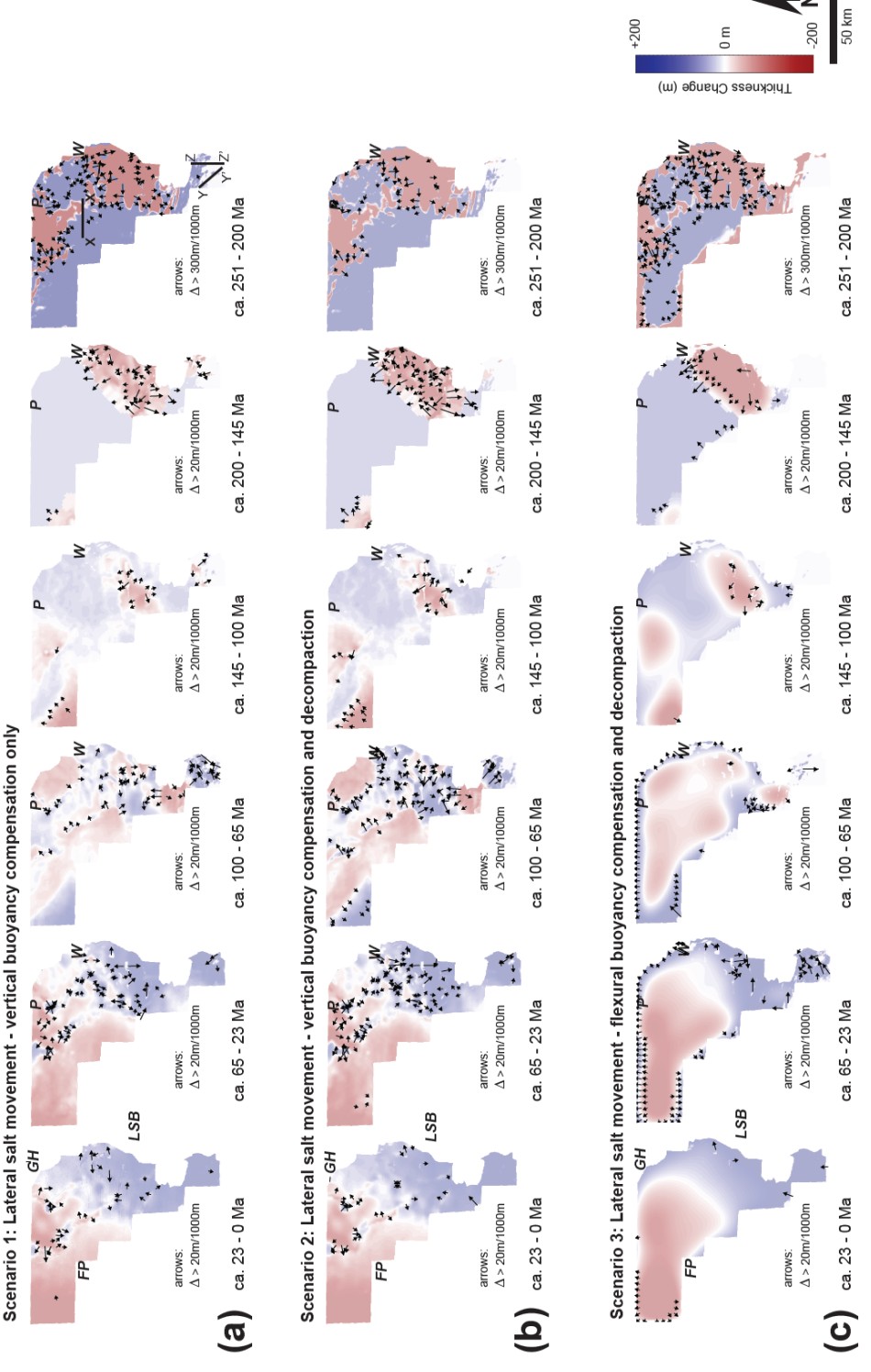

**Figure 8:** Maximum lateral change derived from difference plots of Figure 7. (a) Orientation of maximum lateral change based on backstripping scenario 1. (b) Orientation of maximum lateral change based on backstripping scenario. (c) Orientation of maximum lateral change based on backstripping scenario 3. Note pronounced edge effects at northern boundary of study area associated with flexural backstripping (scenario 3). 2D restoration (scenario 2) extracted along sections X-X' and Z-Z' (see (a), 251 Ma) shown in Figure 5. 2D restoration (scenario 2) extracted along sections X-X' and Z-Z' (see (a), 251 Ma) shown in Figure 5.

The Early Cretaceous (145-100 Ma) shows in all restorations only minor changes in subsurface Zechstein distribution (Figs. 7, 8). Main salt-loss areas are the northern LSB and the eastern FP (Fig. 8). Salt gain is mainly observed in the central and southern LT and along the Hantum fault zone (Fig. 9). The Early Cretaceous (145-100 Ma) of Figure 7c highlights the difference between flexural backstripping and vertical overburden balancing (Figs. 7a, b) in producing a smoothed, partially amplified salt loss and gain plot.

Between 100 and 65 Ma, the GH, LT and eastern FP comprised the main expulsion areas, whereas the LSB and FP locally received >200 m thickness of evaporites (Fig. 7). Vertical balancing with and without decompaction (Figs. 7b, c) documented the growth of two narrow, parallel chains of NW-SE directed salt rollers, anticlines and walls above the main boundary faults of the LT (Fig. 9). These structures received more salt between 65 and 23 Ma (Figs. 8a, b). The LSB accreted in the Paleogene and Neogene on average ca. 200 m of evaporites, respectively, likely sourced from the GH, LT, FP and from regions east (outside) of the study area (Figs. 7, 8). The flexural balancing approach (Figs. 7c, 8c) does not provide sufficient lateral resolution for the determination of Late Cretaceous to recent salt flow into individual salt structures.

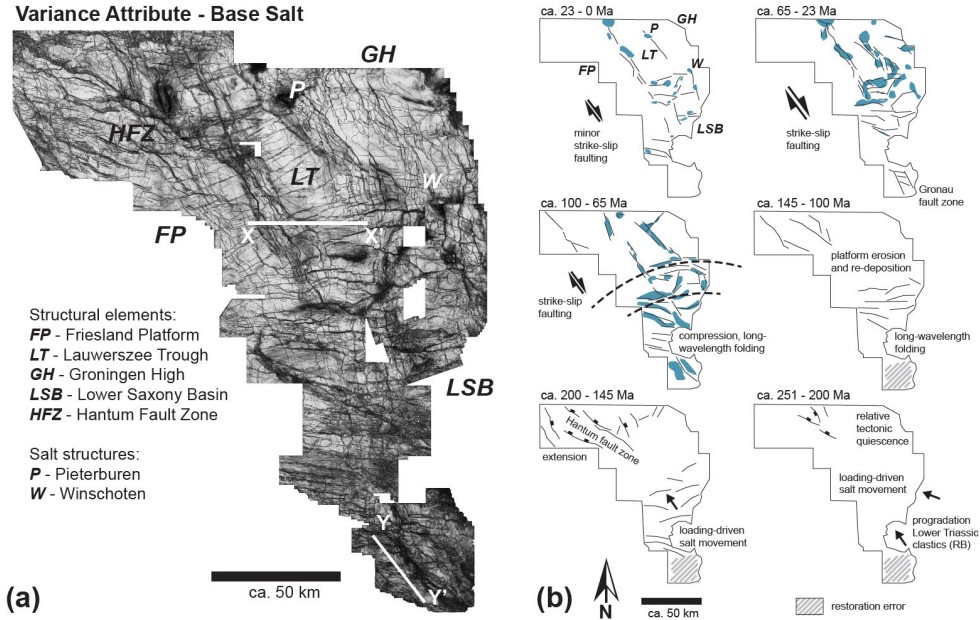

**Figure 9.** (a) 3D variance horizon-slice highlighting main structures of the pre-Zechstein. Hantum Fault Zone (HFZ) between FP and LT characterised by significant salt gain in the Early Cretaceous. (b) Interpretation of relationship between sedimentary processes, tectonics and subsurface salt movement. Seismic-reflection data along lines X-X' and Y-Y' shown on Figure 3. Blue colour indicating main salt structures.

## 4 Geological interpretation

The Zechstein isopach maps (Fig. 6), the evaporite loss-gain calculations (Fig. 7), and the salt-movement plots (Fig. 8) provide important geological information when integrated with tectonic and seismic-stratigraphic analysis (Figs. 3a, b; 5, 9; also see Cartwright et al., 2001; Giles and Rowan, 2012; Alsop et al., 2016; Khalifa and Back, 2021). Salt loss can be interpreted when supra-salt strata forms a thickened overburden. If subsurface evaporites are completely expelled, a salt weld forms. Expulsion forces salt to move elsewhere, and salt either escapes to the surface and dissolves, or flows into salt structures

overlain by an isopach thin, if rising syn-depositionally. Other isopach anomalies, e.g. elongate minima or maxima above basement-rooted structures (Figs. 3a, b; 6-8) can indicate tectonically triggered subsurface salt loss or gain.

        The Zechstein thickness reconstructions document that only small parts of the study area experienced complete salt withdrawal, including a series of large, elongate, mainly E-W-oriented salt welds in the northern LSB (Fig. 8) and parts of the very south of the study area with a lack of reconstructed Zechstein between 200 and 100 Ma (e.g. Figs. 7, 8). The salt-thickness

reconstruction of the south indicates significant salt expulsion and initial diapirism between 251 and 200 Ma; insignificant gain and loss until 65 Ma; and finally salt accretion since 65 Ma (Figs. 7, 9). The pre-65 Ma interpretation of the south must be, however treated with caution. Lack of much of the Mesozoic overburden due to erosion (Figs. 3a, b; 5, 6; 7) results in incomplete Top-Zechstein backstripping. The restored Zechstein base therefore locally intersects the Zechstein top during unloading, producing potential restoration errors (Fig. 7). The locally restored 251 Ma salt thickness of the south (up to 200m)

is yet similar to the salt-thickness reconstruction by Ten Veen et al. (2012).

        Evaporite-thickness change (Fig. 7) divided by the duration of each restoration interval documents that between 200 Ma and present-day, long-term Zechstein thickness change was up to ca. 15 m/Myr. Though low in rate, this change is significant for interpretations on period or epoch scale (Fig. 7). For example, the ca. 150 m growth of salt ridges above the eastern and western boundary faults of the LT in the Late Cretaceous can be interpreted as reflecting overburden thinning due

to inversion tectonics responding to Africa-Iberia-Europe convergence (sensu Kley and Voigt, 2008). The "Airy-type" salt-movement plots between 100 and 23 Ma (Figs. 8a; b) all show significant salt flow above pre-existing, re-activated faults (boundary faults LT; Figs. 3a, b; 9b) into salt diapirs and walls. It must be noted in this context that this regional 3D Zechstein

Basin study did not restore any sub-salt fault movement, which locally limits the accuracy of the thickness reconstruction. However, even if included the post-Triassic evaporite re-distribution will likely remain generally small when compared to the

Zechstein isopach change between 251 and 200 Ma (Figs. 6, 7), which is locally >1500 m (Fig. 7). Long-term evaporite loss (period scale) in the LSB and LT is at this time >30 m/Myr. The Triassic evaporite expulsion can be interpreted as dominantly driven by sedimentary loading from the southeast during the Buntsandstein (duration <10 Myrs; Figs. 2, 3, 5, 6, 9B) in an overall tectonically quiet basin (Mohr et al., 2005; Geluk, 2007; Vackiner et al. 2013; Strozyk et al., 2014). Thus, loading-driven salt flow might attain on epoch scale long-term rates of >150 m/Myr (in line with Zirngast, 1996; Kukla et al., 2008),

which is significantly above the tectonics-driven long-term rate of up to ca. 10 m/Myr in the Late Cretaceous (Fig. 7).

## 5 Discussion

The Zechstein salt system of the NE Netherlands allows regional 3D thickness reconstruction of subsurface evaporites over time; 3D measurement of subsurface salt loss and gain over time; 3D salt-flow reconstruction over time; and the estimation

of long-term salt-flow rates. The reconstruction of 3D subsurface salt movement is – although in this study solely dependent on the restoration of differential overburden thicknesses – not restricted to monitoring sedimentary processes only. Any process that results in differential overburden thickness (including tectonics) will be also be balanced, as exemplarily shown for Late Cretaceous and Paleogene salt-dome growth likely triggered by inversion tectonics (Figs. 7, 8).

Zechstein-thickness restoration enables monitoring the growth and decay of salt structures and salt welds, results that

can be immediately applied in e.g. petroleum systems models or for constraining physical fluid-dynamic models. It must be however noted that the studied Zechstein unit is in nature internally heterogeneous (Fig. 2b), with both vertical and lateral facies variations. More competent lithologies are interbedded with the mobile Zechstein halite units including anhydrite and carbonate stringers (see strong intra-Zechstein reflector, Fig. 3a). Lithological heterogeneity gives rise to rheological heterogeneity, which may have an impact on the associated buoyancy. It must be therefore acknowledged that the assumption

of all Zechstein units as a homogeneous fluid of constant density is a major simplification and thus a likely source of errors.

The Zechstein-thickness reconstructions presented (Figs. 5-8) indicate that likely much of the internal structural complexity of the evaporite succession (for examples see Richter-Bernburg, 1980; Strozyk et al., 2012, 2014; Biehl et al.,

2014) can be explained by recurrent changes between salt loss, salt gain and evaporite-flow direction through time. Although post-Triassic Zechstein thickness change was generally low in rate, it could have well been responsible for the internal

deformation of the Zechstein succession with its various salt layers and intercalated limestones and anhydrites. The slow and rather localized lateral salt flow documented in this Zechstein backstripping study can be seen as key supporting argument for reconstructing subsurface salt flow with a static Archimedean approach.

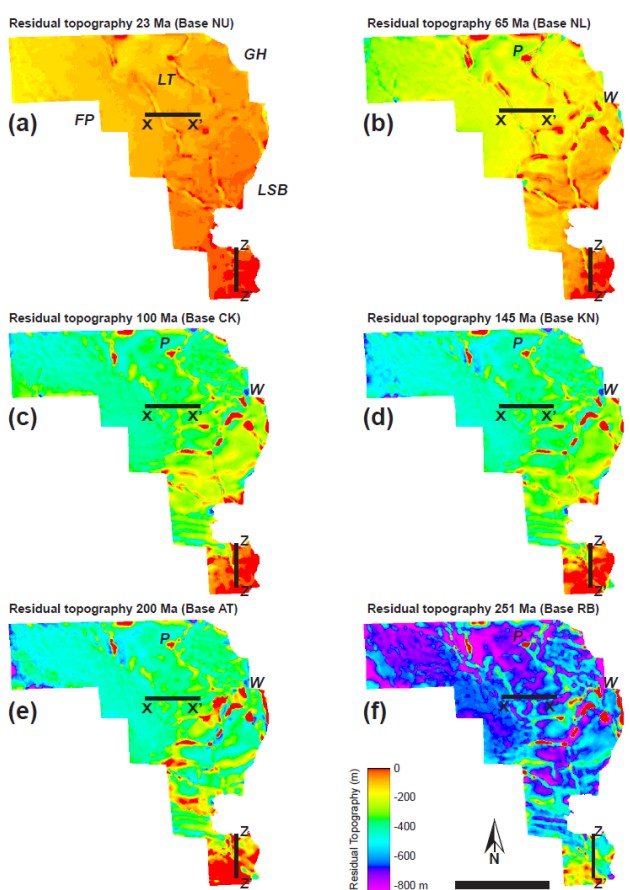

**Figure 10.** Residual 3D topography after each reconstruction step in scenario 2 (unloading and decompaction; submarine conditions). Depth

position of respective model top at (a) 23Ma; (b) 65Ma; (c) 100Ma; (d) 145Ma; (e) 200Ma; and (f) 251 Ma. Note that residual topography does not represent palaeo-bathymetry. Incremental backstripping results in increasingly flat basin-floor topographies away from salt domes and ridges. Note piercement salt domes Pieterburen (P) and Winschoten (W) that remain unbalanced thoughout restoration. For balanced sections X-X' and Z-Z' see Figure 5.

Approximation of the top Zechstein as a horizontal surface at sea level after final unloading (situation at 251 Ma, all

scenarios of Fig. 6) can be potentially used to constrain the 251 Ma depth of the base Zechstein based on overburden unloading

only. In other words, the assumption that the top Zechstein formed at base level roughly approximates the palaeo-depth location

of the base Zechstein, determined from subtraction of the restored 251 Ma Zechstein thickness from the present-day sea level

only. The validity of this approximation yet stands and falls with the validity of the zero-topography assumption for top

Zechstein, as the Archimedean salt restoration presented is not tied to any topographic reference level. Figure 10 shows the

residual 3D topography after each reconstruction step of scenario 2 (i.e. unloading, decompaction, submarine conditions; see

Figs. 5, 6b). The respective topography maps show an increasingly flat basin floor with a remnant topography confined to

unbalanced salt domes and ridges. This configuration can be seen as another argument for the validity of the Archimedean

restoration approach in the Zechstein study example. True palaeo-topographic referencing, potentially providing an alternative

quality control for the differential salt thickness calculation, however can only be achieved by integrating crustal isostatic

balancing above the earth's mantle (Fig. 11; Rowan, 2003; Turcotte and Schubert, 2014) contemporaneously to salt-

redistribution modelling into the restoration process. We have not yet found a technical solution for such an integration; crustal

isostatic balancing remains therefore excluded from this balancing study.

Key limitations of the Zechstein reconstruction presented are the incomplete restoration due to overburden

unconformities; omission of subsalt deformation from restoration; lack of knowledge on loss of salt at piercement structures;

and potential loss or influx of salt at the edges of the study area. Incomplete salt restoration due to overburden erosion primarily

affects the south (Figs. 5, 6; 9b); incomplete restoration of piercement structures various locations. Yet, any backstripping

naturally fails to balance missing rock record unless stratal gaps and hiati are filled. Integration of stratigraphic forward

modelling (e.g. Granjeon, 2014; Grohmann et al., 2021) with backstripping might help closing some unconformable gaps in

the sedimentary record.

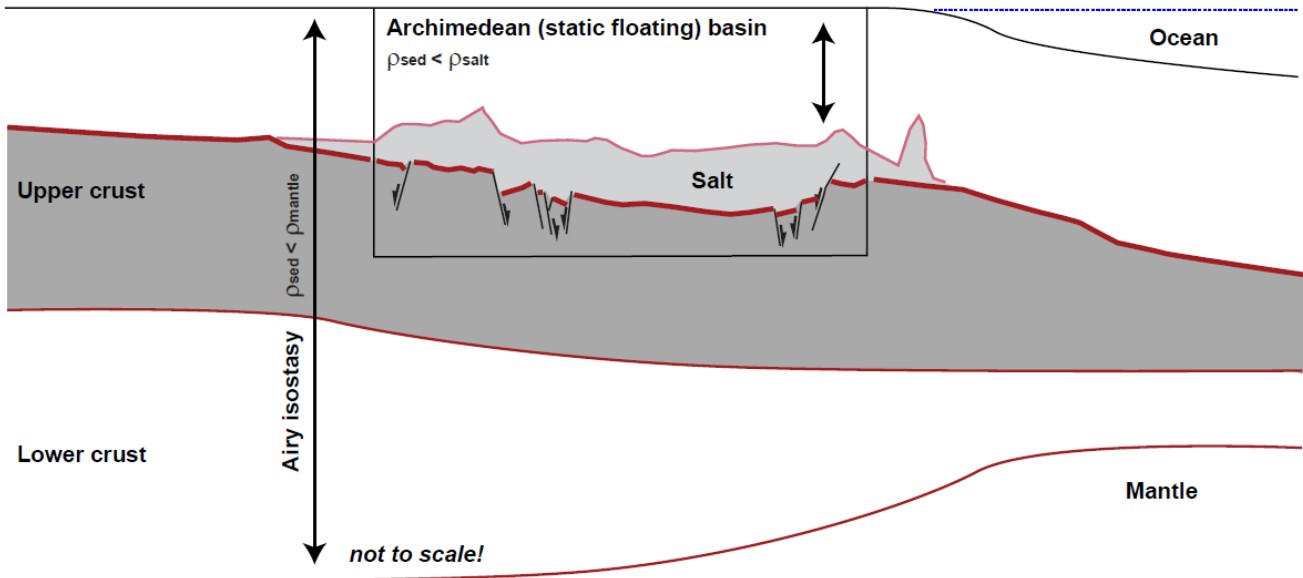

**Figure 11.** Scale and scope of the salt-restoration of this study in comparison to "classic" crustal-scale backstripping (e.g. Turcotte and Schubert, 2014). The concept of isostatic correction is essentially the same for calculating the total mass above a reference datum i) at great depth below the base of the crust; or ii) at Top salt, ignoring everything that happens below that level. Note that volumetrics, lateral distribution, thickness, state of aggregation and physical properties of the respective reconstruction base (salt/evaporites versus the Earth's mantle) are yet fundamentally different.

The salt reconstruction presented has furthermore omitted the restoration of any subsalt post-Zechstein deformation. Yet, if type, timing and magnitude of subsalt faulting or folding can be determined, 3D fault reconstruction or unfolding can be readily integrated into the restoration methodology proposed, in this case after unloading and prior to true-to-volume base Zechstein adjustment.

The magnitude of potential loss or influx of salt at the edges of the study area was finally estimated by comparing a first-pass scenario 1 restoration solely based on 3D block Groningen (Fig. 2) with the scenario 1 restoration of the entire NE Netherlands (Fig. 6a). This comparison showed before 200 Ma a mismatch between the regional and local scenario 1 reconstructions in the Groningen area of <10%. At 200 Ma, the large-scale restoration trailed the local Groningen balance by $0,14 \times 10^{12}$ m$^3$ (ca. 12% of block volume). At 251 Ma, the large-scale reconstruction showed a lack of Zechstein by $0,47 \times$

$10^{12}$ m³ (ca. 40% of block volume) in the Groningen block in comparison to the local model. This imbalance between local and regional reconstructions indicates that true-to-volume balancing highly depends on model size and the amount of differential unloading. Highest precision true-to-volume balancing by unloading will be achieved in restorations that cover subsurface salt systems in full 3D extent; and in these systems in the youngest backstripping intervals.

The case study presented here for the onshore NE Netherlands concentrates on a structurally relatively simple area dominated by vertical subsidence, with limited influence from thick-skinned tectonic activity. The applied method yields in this area promising results. The approach should be equally applicable in other scenarios where a "solid" overburden is less dense than a mobile "fluid" substratum; this potentially includes areas underlain by mobile shale. In scenarios where the overburden reached a cumulative average density above that of the substratum, the unloading methodology can be potentially applied at a later stage in backstripped (restored) former stratigraphic configurations in which an Archimedean equilibrium existed.

The method yet will only work in settings where the salt had enough time to flow so that the sediments and salt could approach Archimedean equilibrium (Fig. 1). In systems where the geology has not yet achieved an equilibrium state the method will not be applicable. For example, if applied to areas where allochthonous salt sheets flow at the surface (e.g. Gulf of Mexico: e.g. Fletcher et al., 1996; Fort and Brun, 2012; Duffy et al., 2019); where complex structures such as salt canopies occur (e.g. Santos Basin: Jackson et al., 2015; Moroccan margin: Neumaier et al., 2016); where large salt nappes have flowed many 10's of kilometres seaward, accommodating long-distance lateral translation of the overburden relative to the base of salt (e.g. offshore Angola; Fort et al., 2004; Hudec and Jackson, 2004); or where sedimentation accumulated rapidly and thick above salt, possibly associated with actively rising salt diapirs, the whole basin system is far from equilibrium and the simple Archimedean method applied here will be insufficient. In such cases a reconstruction coupling 3D salt-thickness restoration and 3D salt tectonic retro-deformation might be successful.

**6 Conclusions**

1. 3D backstripping based on the ancient Archimedes' principle restored through time variations in 3D subsurface evaporite thickness; 3D salt loss and gain; 3D subsurface salt movement; and long-term salt-flow rates.

2. Sequential unloading of a solid sedimentary overburden floating on a pseudo-fluid evaporite substratum showed that subsurface evaporite movement reacts to any process that influences overburden thickness, in this case sedimentation, erosion and tectonics.

3. Limits of buoyancy-based 3D salt reconstruction include incomplete restoration due to overburden unconformities; uncertainty of the volumetric model integrity because of potential salt loss by dissolution; exclusion of subsalt deformation from restoration; and potential loss or influx of salt at the edges of the model area.

4. 3D subsurface salt restoration based on Archimedes' principle is mathematically simple and computationally quick. The approach presented can be potentially integrated into existing backstripping workflows. It can furthermore serve as a

benchmark for physics-based numerical modelling of salt tectonics.

**Data availability.** Access to all seismic-reflection, borehole and geological surface data at www.nlog.nl (Dutch Oil and Gas portal). Lithology data from DINOLoket public database.

**Author contributions.** SB and SA performed data analysis. SB and SA interpreted the data under discussion with VS and RL.

SB, SA, VS and RL wrote the final manuscript.

**Acknowledgements.** We thank the Netherlands Organisation for Applied Scientific Research (TNO) and the Nederlandse Aardolie Maatschappij (NAM) for seismic, borehole and horizon data; Schlumberger for Petrel software; and Petroleum Experts (Petex) for MOVE software. We acknowledge 4 anonymous reviews on an earlier version of this manuscript. Solid Earth reviewer Frank Peel is explicitly acknowledged for his very thorough, constructive and helpful review. Figures 4, 5, 10

and 11 of this paper were drawn following Frank's clear sketches and detailed suggestions. SA was funded by German Research Foundation (DFG) grant 403093957.

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
