# Peer review of "Reconstructing 3D subsurface salt flow"

_Solid Earth, 2021_

## Referee Comment (RC1)

se-2021-153  Review

Reconstructing 3D subsurface salt flow

Stefan Back et al

To the authors:

This is a good piece of work, well worthy of publication. The manuscript as submitted is OK but there are several things that in my opinion need to be improved by way of clarity and by way of content.

My first impression on reading this is that it appears to be an important body of work, but it is very hard to see what is actually going on geologically. The bulk of the explanation is in the form of words of text, plus maps showing salt thicknesses.

The impact of the manuscript as it stands is critically limited by a reliance on maps alone to show the restorations, and on words alone to explain the processes.  I really struggled to understand either the restoration process or the geological results when presented in this way; I believe that the addition of cross sections would help enormously.

The figures in the form of cross sections should be of two types :

(i)     Schematic sections that illustrate the principles – how the restoration process worked, the nature of salt flow, etc.;

(ii)    Selected geological sections through the 3D restoration that illustrate both the sequential evolution of the structure and thickness of salt and the inferred flow of salt through time.

I have included some rough sketches that are all in the form of sections, please excuse the poor quality of these (drawn by hand on an airplane). I hope that these are clear in intent.

My first major point is that the method assumes that the basins are in Archimedean equilibrium (floating) – this is a static equilibrium. However, all actively subsiding withdrawal basins are to some degree out-of-equilibrium, and all actively subsiding basins sit higher than a static floating basin would. This is because the salt has non-zero viscosity. The authors state that a viscous fluid cannot support a shear stress; this is not entirely correct.  Importantly, it only applies to a non-deforming viscous fluid. A fluid that is undergoing shear strain does support a shear stress that is proportional to the viscosity and the shear strain rate. The surfaces of the cargo carrier in your Figure 1a will experience significant shear stress (water resistance) as soon as the ship starts to move.

IS THE ARCHIMEDEAN ("FLOATING")
MODEL APPLICABLE TO WITHDRAWAL BASINS?

FLOATING (AIRY ISOSTATIC)

SEDIMENT

SALT

datum equal pressure

SALT PRESSURE AT DATUM
IS CONSTANT SPATIALLY
→ NO HALOSTATIC GRADIENT
→ NO LATERAL SALT FLOW

ACTIVELY WITHDRAWING BASIN

↓SEDS↓

HIGH → LOW
pressure head

• SALT IS FLOWING
• HEAD (POTENTIAL) GRADIENT
  MUST EXIST
• CANNOT BE IN IOSTATIC EQUILIBRIUM
  ACROSS THE MODEL

The manuscript needs to recognize and discuss this. You should explain what difference it would make, and discuss why it might be a reasonable approximation to use a static Archimedean (floating) model – I would suggest that the primary justification might be that lateral salt flow is very slow in this region, particularly when compared to withdrawal basin systems such as seen in the Gulf of Mexico or Pricaspian, so the non-static stress supported by the salt is less of an issue. But this is for the authors to decide.

The explanation of the restoration method is inadequately described. This is another case where explanatory sketches would be a huge help. I had to guess what the process you use is, and here is my rough sketch.

[Figure]

FOR ARCHIMEDEAN ("FLOATING") MODEL
(WITH NO SEDIMENT RIGIDITY)....

$$\text{weight of sed} \atop \text{+salt above} \atop \text{deep ref. level} = T_{sw} \times \rho_{sw} + T_c \rho_c + T_D \rho_D + T_{SALT} \times \rho_{SALT} = W$$

W is constant value laterally across
the model

It would also help to explain the difference between a standard vertical-shear reconstruction, and the Archimedean approach used here. Again, here is my rough sketch.

[Figure]

Because you do not explain what the process is, I had to guess: this is my best guess. In a standard vertical-shear restoration, the top surface is restored to either a flat horizontal surface, or to some estimated paleotopography. It sounds like, in your method, the top surface is defined by making each vertical column of sediment "float" on the salt, - and as a result there is a residual topography on the top surface of the restoration. This top-surface topography seems to be a necessary part of an Archimedean restoration. If there is not top-surface topography in the restoration, it is simply a flattening not an Archimedean restoration. While we can accept that you do not constrain the absolute elevation of the restored top surface, it would seem necessary that your model does give a prediction of relative topography of the restored top surface. This is an important point, and one that you need to make in the paper. I would like to see a discussion about the amount and shape of the top surface topography that appears in your restoration. If it turns out that after you do the full Archimedean restoration, the calculated restored top-surface topography appears to be near to horizontal, this is an important point to make, because it would be strong evidence that the method is valid.

Further, you need to state what you used for the density of the space above the top of the sediment pile in the restoration . Did you assume that it is seawater or air?

Please forgive me if I have misunderstood the nature of your restoration process. If I have misunderstood, it would show that the description provided in the manuscript is not clear or complete.

Another important point that is glossed over in the paper is that the restoration process for the suprasalt may be independent of any basement tectonics affecting the subsalt. It may also be independent of any imposed thin-skinned translation of the cover, as long as the cover is neither shortened nor thickened.

But your quantified analysis of inferred salt flow (which derives from your restoration) is NOT independent of subsalt deformation. The very rough sketch below illustrates this point.

[Figure]

In this sketch, a base-salt offset is created by an inverted rift fault. If the section is restored without also taking into account the movement of the basement (top panel) then a large volume of lateral salt flow would be required. Conversely, if the basement faulting is restored (bottom panel) then the amount of lateral salt flow may be minimal. This example is used because the effect of basement tectonics is clearest, but the same principle will apply to any movement of the basement that is not uniform upward/downwards motion – even a gentle regional tilting of the basement would change the apparent salt flow.

The presentation of results is entirely in the form of maps. These are OK as far as they go but they do not do justice to the work that has been done; they need to be supplemented with a set of good cross sections from the restoration that show the subsalt, salt and suprasalt layers, and which illustrate the sequential evolution of the section through time. This is standard practice for a very good reason; it is easy to see how the geology evolves in cross section, but if you only show maps of salt thickness, it does not give an understanding of how the salt thickness relates to the suprasalt geology. If a 3D restoration has been carried out, it should be a simple matter to extract the cross sections from this. The cross

section lines should be chosen so as to illustrate the main results of the salt flow analysis, and the flow direction of the salt at each time should be shown on the sections. This will greatly increase the intelligibility and impact of the manuscript.

Page 3 lines 60 onwards – wrong use of Early – Mid – Late (chronostrat terminology) vs. Lower – Middle – Upper (lithostrat terminology) . Lower Cretaceous rock units, Early Cretaceous deposition.

Page 4  - this study treats the Zechstein as one evaporite unit behaving as a fluid . How reasonable an approximation is this, given that the Zechstein consists of a layered sequence which includes thick, competent non-evaporite units? Discuss and justify.

P4 lines 75-80 need to state whether 3D data sets are in TWT or in depth

P5 figure. Vertical axis is NOT depth. Relabel as TWT (mS). Note capital S in mS.

P6  line 90 evaporite not evaporate

P6  explain density/depth model better. What does "density" in Table 1 mean – is it density of material at surface (i.e. with initial porosity), or density when porosity is reduced to zero? What density do you assume for the pore-filling fluid? (we assume that you do include the weight of the  pore fluid in the weight calculation but this is not specified).

P 11. You need to be clearer about what salt gain or loss mean. My interpretation of what I think you mean is that it is only the local gain or loss, representing the lateral flow of salt within the model. If  I am reading this correctly, you should state that salt loss = local salt withdrawal and lateral expulsion; salt gain = region of local salt inflation by lateral influx.

P12

 lines 160 onwards – unclear wording, especially line 162. Replace "top salt" with "suprasalt".

P13.

Line 165 v unclear, rephrase.

Lines 166- 170 unclear.

line 169-70 – this is the first mention of allochthonous salt. If allochthonous salt sheets are present, the whole reconstruction model would be invalid. Also, do you reallly mean allochthonous salt? I am not aware of significant allochthonous salt bodies being present in this region.

Line 180 – this refers to the movement of salt above reactivated subsalt faults. But as discussed above, if there is significant movement of any fault below the base salt, this would invalidate the reconstruction logic.

P14

Line 191 – we don't need to know about Archimedes

Line 207 – suggest delete the sentence "spatial evaporite…. "

P 15

Line 235 there is not a "lack" of Zechstein. Suggest replace "lack" with "volume deficit" or something similar

P16

Lines 243-245 these three clauses do not describe different geometries, they are essentially the same thing. Salt canopies ARE  allochthonous salt sheets and big sheets may be called nappes.

---

## Author Comment (AC2)

**Table 1:** Stratigraphy and average rock properties (after Hunfeld et al. 2021) used for backstripping and decompaction.

[revised manuscript text omitted]

---

## Author Response (AR1)

Response to the comments made by Frank Peel (referee #1)

Dear Frank,

first of all thank you very much for all the work you put in reviewing "Reconstructing 3D subsurface salt flow" (se-2021-153)! We have rarely received such a through, informative and constructive review! Also many thanks for your sketches. These are super informative! We have now submitted a revised document in which we incorporated all suggestions from your side.

Here the response to your comments:

1. "Addition of cross sections to the paper" (your page 1):
   Please find in the attached pdf a revision of all figures – now including 4 cross-section figures directly based on your cartoons.
   - Figure 1 now includes two cross sections (b and c) that illustrate the difference between a basin in Archimedean equilibrium and an actively withdrawing supra-salt basin. We have furthermore added some simple formulas for further understanding. We will use this figure in the revised document to better discuss why we use the static equilibrium model for our salt reconstruction (also see point 2).
   - The cross sections of new figure 4 shows the restoration procedure of this paper, guided by sketch number 2 of your review. It is shown that the backstripping approach only includes unloading (and for scenarios 2 and 3 decompaction); this contrasts the classical vertical shear methods for palaeotopographic restoration.
   - The cross sections of new figure 5 are 2D restorations extracted from the 3D restoration along seismic lines Norg XL 8000 and Twente IL 9000 (location on figures 2, 6, 7, 8, 9, 10).
   - The cross section of new figure 9 shows the difference between the restoration approach of this paper and classic Airy-isostatic balancing based on a figure you sent informally by email.

2. "Applicability of Archimedean equilibrium approach" (your pages 1 and 2):

   We adressed this important point in the revised ms in the two last paragraphs of the introduction. Revised Figures 1b and 1c (see above and attachment) now allow to show and discuss the difference between a basin in static equilibrium and one which is not. You suggested that salt withdrawal and lateral salt flow in the Zechstein basin was rather small (in comparison e.g. to the GOM) and therefore likely allowed application of the equilibrium model; we now discuss this, and point out at the same time that the model procedure forwarded is also in the study area limited, e.g. when reaching piercement structures or in the very initial phase of post-salt sedimenttation (Early Triassic).

3. "Better explanation of restoration procedure" (your page 4):

   We now provide an improved description of the restoration procedure based on new figure 4. We added detail to the originally too brief method section; e.g. your question about "the space above the top surface: air or seawater". We originally restored everything subaerially. Following our informal email exchange and looking at the restored top surfaces (residual topographies) of the model (-> please see new figure 10; might be shifted in place in the revised ms), we ran all restorations again using submarine conditions (seawater density above restored top surface).

Remodelling was anyway necessary because of the use of wrong physical property values for the Chalk Group (see referee#2).

4. "Restoration process independent of basement tectonics":

We hopefully clearifed this in the revised paper. It is clear that any change in basement configuration will affect the current model; however, if known, where and what kind of change occurred, this can be introduced into the restoration procedure.

5. All comments on page 6:

All followed.

What we did not do: We left the TWT unit as "ms TWT" (SI unit) -> Fig. 3

In summary, thank you very much for your very thorough and helpful review! Your review provided the base for 3 completely new figures, and one completely revised figure. The new residual topography figure 9 also results from a suggestion by you (and your group in your emails).

Best wishes

Stefan et al.

Response to the comments made by referee #2

Dear referee #2,

thank you very much for your straightforward, constructive and positive review of the ms "Reconstructing 3D subsurface salt flow" (se-2021-153).

Here is our response to your comments:

(1) "Introduction, motivation for conducting the study can be addressed in a clearer way":
   We addressed the motivation for conducting the study more clearly. We stressed that the method proposed provides insights into 3D subsurface salt flow and redistribution on basin-scale, the rise and fall of salt structures and associated depocentre development, and external forces' impact on subsurface salt movement.

(2) "Table 1, density", but also Young's Modulus, Poisson Ratio of Chalk:
   The physical properties for the Chalk were wrong. This error was not only in table 1, the same mistake was also in the lithological model used for backstripping and decompaction. We consequently re-ran all restorations of the study with revised chalk values (from onshore NL) provided by Hunfeld et al. (2021). Please find in the attached pdf the revised table and revised model results (e.g. in figures 5; 6; 7; 8; and 10). Please note that the revised models additionally contain a change suggested by reviewer Frank Peel: surfaces that were restored to a level below zero were treated as submarine, contrasting the original model (fully subaerial restoration). Yet, the new model results are quite similar to the original restoration results.

(3) "present-day cumulative average density":
   Yes, this is grain density + porosity; we will make clear in the upcoming revisions.

All other suggested changes (lines 129, 144, 145, 146-147, 170-173, 210, 212) are now in the revised manuscript.

Again, thank you very much for your helpful review, particularly for exposing the mistake in our lithological model!

Best wishes

Stefan et al.

Reference: Hunfeld, L.B., Foeken, J.P.T., and van Kempen, B.M.M.: Geomechanical parameters derived from compressional and shear sonic logs for main geothermal targets in The Netherlands. TNO: https://www.nlog.nl/sites/default/files/2021-12/data_selection_and_methods.pdf, last access: 11.04.2022.

---

## Referee Report (RR1)

Abstract –

This could state that the paper presents an approach to palinspastic restoration that is radically different from from the usual methods.

Introduction (line 75)

I think that you need to say more about the geological setting and structural style of the salt here.

To introduce the salt structures

It's significant that there do not appear to be any diapirs or salt-withdrawal basins in the region (at least as far as I can tell from your maps and sections) – this could be invoked as supporting evidence for the statement that the sediments remain less dense than salt.

It's also significant that there are salt structures, and that these salt rollers, anticlines and walls appear NOT to be diapiric, instead they are the product of lateral shortening. This again should be stated, because it's important.

[Figure]

present day

restoration

And thirdly, it looks like many of the compressional salt rollers, anticlines and walls did not grow further when they were buried, and indeed, it looks like the _opposite_ happened; many of the early salt-cored highs saw the salt flow OUT of the highs, as illustrated by line X-X' (here is a quick and dirty reconstruction that illustrates the point). Again, I think this is important to say, because it also provides supporting evidence that the cover section was less dense than the salt.

I suggest that it is really important to describe what sort of salt structures you are dealing with before launching in to how you restored them.

Sediment densities and compaction (line 105)

The statement that the _average overburden density remains lower than that of salt in all cases_ is surprising, and this probably needs more justification. My own research on this subject has indicated that many common sediments are deposited with density close to that of salt (e.g. most platform carbonates, most "dirty" continental clastics such as typical fluvial deposits ), and that those that are

substantially less dense than salt at time of deposition tend cross over in density at about 1km of burial. The density crossover in most basins appears to be at around 1km or shallower. This scenario has been described in several publications. You may well be right about your study area being different, and always maintaining average density lower than salt, but you should note that it appears to be an exception to the most commonly described situation in other basins.

I think that you are correct in your study area, and I suspect that the two main reasons why your study area has this property, (which is unusual on the global stage), are that:

1. The Chalk is an unusual (strong and low density) rock – it is a limestone with extremely high initial porosity (thus extremely low initial density compared to "normal limestone") and this porosity can be preserved at depth if the pore fluids are sealed in. For example, the high initial porosity in the Ekofisk Field was preserved (with 30-50% porosity at depth),  because the pore fluid could not escape until the field went on production – at which point the rock fabric microstructure collapsed and the Chalk compacted.
2. The depth of burial has never been very great (<2km or so total, chalk burial <1km). Not great enough to collapse the Chalk porosity.

The other factor that you should mention at this stage is to quantify the typical thickness of the suprasalt sediment section in your study area. EVERY type of sediment will become more dense than salt at some depth! If the typical thickness was <1km or so, then it would not be surprising that the sediment is less dense than salt. But given that the total thickness of sediment above salt is about 2km on line X-X', (fig 5), I was quite perplexed, and I suspect that so will be many of the readers.

Perhaps a simple density/depth plot would be helpful.

[Figure]

The lithologies used need clarification, and it would be good to list the % sand. When you write that the Upper North Sea Group is sandstone, are you saying that it is 100% sand, or that it is (say) 60% sand/40% shale?

Can you state whether the density/depth relationships were taken from Sclater and Christie without modification, or whether they were cross-checked against the density indicated by wireline log data in the study area?

At the end of the discussion, I think that it would be valuable to add a paragraphs that compare/contrasts your method with conventional restoration, and a paragraph that discusses where the method might be applicable and when it might not.

I believe that your method could apply equally well for scenarios where the sediment is less dense than the salt, and in scenarios where the sediment is denser than the salt. This is worth saying, because otherwise the reader might interpret that the method only applies where the sediments are less dense, as in your study area.

However, the method only applies where the salt has had enough time to flow so that the sediments and salt can approach Archimedean equilibrium (as shown in b and d in the sketch below). In situations where the geology has not yet achieved Archimedean equilibrium (a and c , in the sketch below) the method will not be applicable. I suggest that a near-equilibrium scenario can be achieved where the rate of sediment accumulation is relatively slow (e.g. in the North Sea, and in your study area, or in the Pricaspian Basin of Kazakhstan). But in regions where the sediment accumulation rate is very rapid (e.g. the Cenozoic Gulf of Mexico, Kwanza and Congo basins of Angola, Santos Basin of Brazil) the basins are actively subsiding, the salt diapirs are actively rising, and the whole ensemble is far from equilibrium. In such settings, I would not recommend the use of your method.

**sediment density > salt**

non Archimedean
not in static equilibrium
salt is actively flowing
sediment is moving vertically

Archimedean
In static equilibrium
salt is not flowing
sediment is not moving vertically

[Figure]

[Figure]

**in a and b, sediment is denser than the salt**

**sediment density < salt**

[Figure]

[Figure]

sediment

salt

**in c and d, sediment is less dense than the salt**

---

## Author Response (AR2)

Dear Frank, dear Federico,

thank you very much for your comments and suggestions for the ms "Reconstructing 3D subsurface salt flow". We have considered all comments and worked on the respective parts in the text. Please find below a detailed description of the changes made, indicated by arrows. Please find furthermore one version of the text and figures uploaded with track changes, and a final document uploaded with all changes accepted. All corrections marked in the annotated ms were made.

Revisions:

1. "Abstract – This could state that the paper presents an approach to palinspastic restoration that is radically different from from the usual methods."

-> lines 19-21: The 3D reconstruction procedure is radically different from classic backstripping in limiting palinspastic restoration to the salt overburden; followed by volume-constant balancing of the salt substratum.

2. "Introduction (line 75) - I think that you need to say more about the geological setting and structural style of the salt here. To introduce the salt structures (…)"

-> now introduces more about the geological setting and the type of salt structures: The Permian Zechstein Group in the subsurface of the Netherlands, Central Europe (Fig. 2a) accumulated in the foreland of the Variscan orogen (Geluk 2007). The Zechstein Group of the onshore Netherlands comprises five evaporite cycles (Z1-Z5; Van Adrichem Boogaert and Kouwe, 1993; Geluk, 2007) with several hundreds of metres of rock salt and anhydrite deposited mainly in Z2 and Z3 (Fig. 2b). Several small tectonic pulses are reported to have occurred during Zechstein times, with partly extensional and partly compressional faulting mainly affecting anhydrite platforms at the Zechstein Basin margins (Geluk, 1999). The occurrence of Zechstein evaporites in the Netherlands' subsurface influenced the post-Permian geological development. The visco-plastic behaviour of salt under loading and compressive tectonic stress (Remmelts, 1995) led to the development of numerous salt structures, mainly salt rollers, salt anticlines and salt walls (e.g. Fig. 3). Many of these structures were not actively diapiric and did not grow further when buried (e.g. Trusheim, 1963).

-> Caption figure 3 additionally mentions type of salt structures

3. "Sediment densities and compaction (line 105)"

-> revision of significant parts of the text. We have not added an additional figure with depth, porosity, and/or density trend; figures A1a, A1b and A2 of Sclater and Christie (1980):

[Figure]

Fig. A1a. Simplified plots of log porosity versus depth for shales and sandstones. The North Sea shale data were computed from the observed sonic log plots for a normal pressured section of Schlumberger [1974] using the sonic log velocity/porosity relation of Magara [1976a]. The sandstone data for the North Sea are from Sealey [1976] supplemented with data calculated using observed sonic velocities and the velocity/porosity tables of Schlumberger [1974]. The best fit straight line through the data and that through the Atwater and Miller [1965] data is constrained to pass through the surface porosity values of Pryor [1973].

[Figure]

Fig. A1b. Plots of log porosity versus depth for chalks and shaley sands in the North Sea. The chalk data are from Scholle [1976] supplemented by porosity values in two normally pressured holes, calculated using the sonic logs and the velocity/porosity relations of Schlumberger [1974]. The best fit straight line through the data is constrained to pass through the DSDP data and the lowest values of the two chalks in the northern portion of the Central Graben. The porosity of the shaley-sand used to estimate compaction in sections below the Late Cretaceous was taken as the average of the relations for sand and shale individually.

[Figure]

Fig. A2. Estimated porosity versus depth relation for Amoco 2/11-1. The relation was determined assuming that total sealing occurred at the end of the Eocene and with normal pressure above.

as these show in a very clear way the porosity versus depth relation for their template lithologies used (without modification; now explicitly stated – see below) in this study.

-> We have however integrated the explicit mentioning of high chalk porosity but also limited burial of the sand-prone North Sea Group as suggested by Frank. The revised text is now: "Strata above the Zechstein were assigned average lithologies (Table 1) with the definition of average rock type (shale, sand, chalk, shaley sand), compaction trends and density/depth relationships taken from the North Sea database of Sclater and Christie (1980) without modification. Young's Modulus and Poisson Ratio data are from Hunfeld et al. (2021). In all cases the present-day cumulative average density of the column of vertical overburden (= grain density + porosity; pores filled with water) was lighter than the density of the evaporite substratum (fluid with $\rho$ = 2.2 g/cm3), and should have been so in the past.

The backstripping observation that the cumulative average overburden density remained in the study area always lower than that of the Zechstein Group might be surprising, as every sediment will become at some depth denser than salt. Yet, in the study area the depth of Top Zechstein was never very great (in most areas < 2500 m; see Fig. 3 for present-day situation). Since i) both the Chalk Group and the North Sea Group were never deeply buried; ii) both groups constitute the main part of the overburden (Fig. 3); and iii) chalk can preserve very high porosities at depth (30-50% at ca. 2500m in the North Sea example of Sclater and Christie, 1980), we estimate that in the study area over 3000 m of sedimentary cover with a significant shale content would be needed to attain a cumulative average overburden density exceeding 2.2 g/cm3."

-> going back into the Sclater and Christie (1980) depth-porosity curves with the average lithology densities and pores filled with water, burial of > 2500m would be needed to get cumulative average density of the sediment column > 2.2 g/cm3.

4. At the end of the discussion, I think that it would be valuable to add a paragraphs that compare/contrasts your method with conventional restoration, and a paragraph that discusses where the method might be applicable and when it might not.
-> We have re-written the two last paragraphs of the discussion. We didn't include a comparison with conventional restoration – this is already mentioned in the methods section; and in lines 294-298 with the reference to figure 11. We however state, as suggested, that the method could apply both for scenarios where the sediment is less dense than the salt, and in scenarios where the sediment is denser than the salt. In the latter case, however, the method can only be introduced at a later stage in the restoration. With our simple loading model, a too dense overburden column will simply fall through a too light substratum (with cumulative average overburden density greater than that of the substratum).

-> The last paragraph explicitly states where and why the method might not work.

-> The revised text is now: "The case study presented here for the onshore NE Netherlands concentrates on a structurally relatively simple area dominated by vertical subsidence, with limited influence from thick-skinned tectonic activity. The applied method yields in this area promising results. The approach should be equally applicable in other scenarios where a "solid" overburden is less dense than a mobile "fluid" substratum; this potentially includes areas underlain by mobile shale. In scenarios where the overburden reached a cumulative average density above that of the substratum, the unloading methodology can be potentially applied at a later stage in backstripped (restored) former stratigraphic configurations in which an Archimedean equilibrium existed.

The method yet will only work in settings where the salt had enough time to flow so that the sediments and salt could approach Archimedean equilibrium (Fig. 1). In systems where the geology has not yet achieved an equilibrium state the method will not be applicable. For example, if applied to areas where allochthonous salt sheets flow at the surface (e.g. Gulf of Mexico: e.g. Fletcher et al., 1996; Fort and Brun, 2012; Duffy et al., 2019); where complex structures such as salt canopies occur (e.g. Santos Basin: Jackson et al., 2015; Moroccan margin: Neumaier et al., 2016); where large salt nappes have flowed many 10's of kilometres seaward, accommodating long-distance lateral translation of the overburden relative to the base of salt (e.g. offshore Angola; Fort et al., 2004; Hudec and Jackson, 2004); or where sedimentation accumulated rapidly and thick above salt, possibly associated with actively rising salt diapirs, the whole basin system is far from equilibrium and the simple Archimedean method applied here will be insufficient. In such cases a reconstruction coupling 3D salt-thickness restoration and 3D salt tectonic retro-deformation might be successful.

In summary, we are very very grateful for the thoughts, comments and suggestions provided by Frank Peel! We see these as a major contribution to the manuscript, ultimately leading to a more accurate scientific document with broader significance. Thank you very much! Otherwise we hope that the ms is now in an accepteable form; we highly appreciate the professional editorial handling of the ms!

Best wishes

Stefan et al.

---

## Author Response (AR3)

Author's response:

Colour schemes - I am red-green blind myself (completely red blind and weak with green); colour identification additionally checked with colleagues with colour vision deficiency.